



# Reducing Risk Together: moving towards a more holistic approach to multi-(hazard-)risk assessment and management

Philip J. Ward[1,2], Sophie Buijs[1], Roxana Ciurean[3], Judith Claassen[1], James Daniell[4], Kelley De Polt[1,5], Melanie Duncan[6], Stefania Gottardo[7,8], Stefan Hochrainer-Stigler[9], Robert Šakić Trogrlić[9], Julius Schlumberger[1,2], Timothy Tiggeloven[1,8], Silvia Torresan[7,8], Nicole van Maanen[1], Andrew Warren[2], Carmen D. Álvarez-Albelo[10], Vanessa Banks[3], Benjamin Blanz[11], Veronica Casartelli[7,8], Jordan Correa González[10], Julia Crummy[6], Anne Sophie Daloz[12], Marleen C. de Ruiter[1], Juan José Díaz-Hernández[10], Jaime Díaz-Pacheco[10], Pedro Dorta Antequera[10], Davide Ferrario[7,8], Sara García-González[10], Joel Gill[13], Raúl Hernández-Martín[10], Wiebke Jäger[1], Abel López-Díez[10], Lin Ma[12], Jaroslav Mysiak[7,8], Diep Ngoc Nguyen[7,8], Noemi Padrón Fumero[10], Eva-Cristina Petrescu[14], Karina Reiter[9], Jana Sillmann[11], Lara Smale[6]

[1]Vrije Universiteit Amsterdam, De Boelelaan 1111, 1081 HV Amsterdam, the Netherlands
[2]Boussinesqweg 1, 2629 HV Delft, the Netherlands
[3]British Geological Survey, Nicker Hill, Keyworth, Nottingham, NG12 5GG, United Kingdom
[4]Risklayer, Bismarckstraße 59, 76133 Karlsruhe, Germany
[5]Max Planck Institute for Biogeochemistry, Hans-Knöll-Str. 10 07745 Jena, Germany
[6]British Geological Survey, The Lyell Centre, Research Avenue South, Edinburgh, EH14 4AP
[7]Venice Ca' Foscari University, Edificio Porta dell'Innovazione, 2nd floor, Via della Libertà 12, Venice, 30175, Veneto, Italy
[8]CMCC Foundation - Euro-Mediterranean Center on Climate Change, Edificio Porta dell'Innovazione, 2nd floor, Via della Libertà 12, Venice, 30175, Venice, Italy
[9]International Institute for Applied Systems Analysis (IIASA), Schlosspl. 1, 2361, Laxenburg, Austria
[10]University of La Laguna (ULL), Calle Padre Herrera, s/n, 38200 San Cristóbal de La Laguna, Tenerife, Spain
[11]University of Hamburg, Grindelberg 5, 20144 Hamburg, Germany
[12]Center for International Climate Research (CICERO), Postboks 1129 Blindern, 0318 Oslo, Norway
[13]Cardiff University, School of Earth and Environmental Sciences, Cardiff University, Main Building, Park Place, CF10 3AT, Cardiff, UK
[14]Bucharest University of Economic Studies, Piata Romana 6, Sector 1, 010374 Bucharest, Romania

*Correspondence to*: Philip J. Ward (Philip.ward@vu.nl)

**Abstract.** Moving towards a more holistic approach to disaster risk management, in which a multi-(hazard-)risk approach is central, offers many opportunities to increase society's resilience. In 2022, we presented a research agenda of six points that could contribute towards this paradigm shift. In this paper we synthesise key learnings from the MYRIAD-EU project - which ran from September 2021 to December 2025 - reflecting on progress and challenges faced in pursuing this research agenda, and share perspectives that may help to further improve multi-(hazard-)risk assessment and management. Going forward, we point to several avenues for continued scientific research: continue the mainstreaming and mutual understanding of concepts and definitions; continue developing a strong evidence base of how multi-(hazard-)risk both shapes, and is shaped by, risk dynamics over space and time; further developing methods for providing both current and future multi-



(hazard-)risk scenarios; increasing the availability of appropriate, solutions-oriented, usable tools; more explicitly including
equity issues and equitable disaster risk reduction and adaptation; continue extensively testing and coproducing multi-
(hazard-)risk knowledge in in-depth case studies; supporting the development of Multi-Hazard Early Warning Systems; and
strengthening opportunities for Early Career Researcher leadership and empowerment within project structures. We suggest
concrete ways to advance on these topics in future years and decades.

## 1 Introduction

Effective Disaster Risk Management (DRM) is essential for achieving a sustainable future. In 2015, the Sendai Framework
for Disaster Risk Reduction 2015-2030 was launched, calling for "*The substantial reduction of disaster risk and losses in
lives, livelihoods and health and in the economic, physical, social, cultural and environmental assets of persons, businesses,
communities and countries*" (UNDRR, 2015, p. 12). The recent Mid-Term Review (MTR) of the Sendai Framework states
that progress towards its overall goal has stalled and, in some cases, reversed. Specifically, it states that there is an "*...urgent
need to adopt multi-hazard risk reduction approaches that address all risks at source before they manifest as shocks or
disasters...*" (UNDRR, 2023, p. 23). However, it also clearly states that there is currently a significant gap in our ability to
comprehend and manage these risks (UNDRR, 2023), as also discussed by Tiggeloven et al. (2025a).

Within the EU, the importance of this multi-hazard approach to DRM has taken centre stage in the policy arena. The EU
Preparedness Union Strategy was recently set up to boost the EU's ability to anticipate, prevent, and respond to these
complex risks (European Commission, 2025). The strategy calls for an "integrated all-hazards approach", focusing on
preparing for and responding to all types of hazards, rather than addressing them separately. Moreover, the recent update of
the UNDRR/International Science Council Hazard Information Profiles (HIPs) (UNDRR & ISC, 2025) includes a multi-
hazard context.

To develop the scientific knowledge required to support this, over the last five years the EU has funded several large
research projects on the challenge of multi-(hazard-)risk assessment and management, including: MYRIAD-EU, PARATUS,
The HuT, MEDiate, MIRACA, ANTICIPATE, C2Impress, COMPASS, and TOGETHER. Each of these projects focuses on
a different aspect of the challenge, with all projects interacting to collectively try to increase our understanding of multi-
(hazard-)risk.

Ward et al. (2022) set out a research agenda outlining six key challenges that need to be addressed in order to help us better
understand and manage multi-(hazard-)risk. The research agenda called for the following:

- Establishing a set of common multi-(hazard-)risk definitions and concepts and an overview of existing methods and
  tools;
- Co-developing a framework for multi-(hazard-)risk assessment and management;
- Increasing understanding of dynamic feedbacks between hazard, exposure, and vulnerability;
- Developing future scenarios of plausible multi-(hazard-)risk;



- Assessing the effectiveness of DRM measures across hazards, sectors, and time horizons;
- Testing of approaches in in-depth case studies on multi- (hazard-)risk assessment and management.

Each of these was tackled by the MYRIAD-EU project (Sept 2021 - Dec 2025). In this paper we synthesise the key learnings, reflecting on progress and challenges faced in pursuing this research agenda (Section 2), and set out a forward-
looking perspective on how to further catalyse the paradigm shift towards a more holistic approach to multi-(hazard-)risk assessment (Section 3).

## 2 Key learnings

The MYRIAD-EU project had the overall aim to enable policymakers, decision makers, and practitioners to develop forward-looking DRM pathways that assess tradeoffs and synergies of various strategies across sectors, hazards, and spatial
scales. At its core were five Pilots: North Sea, Canary Islands, Scandinavia, Danube Region, and Veneto. Within each of these Pilots, we worked together with multiple stakeholders to co-produce and iteratively test methods, tools, and approaches for assessing and managing multi-(hazard-)risk. Each Pilot covered a selection of three interlinked sectors, from ecosystems & forestry, energy, finance, food & agriculture, infrastructure & transport, and tourism. In this section, we present key learnings related to each of the research agenda points (Sections 2.1-2.6), highlighting key challenges encountered along the
way. We also briefly reflect on several other key learnings that were gained throughout the course of the project (Section 2.7).

### 2.1 Establishing a set of common multi-(hazard-)risk definitions and concepts and an overview of existing methods and tools

We undertook research with the objective of building a common baseline of terminology and concepts, and academic,
policy and industry perspectives, on multi-(hazard-)risk management. This challenge was approached through three complementary activities: (1) the collaborative establishment of a set of common multi-(hazard-)risk concepts and definitions, and indicators for use throughout MYRIAD-EU; (2) a comprehensive review of existing qualitative and quantitative methods, models, and tools relating to multi-(hazard-)risk management; and (3) a review of policies, policy-making processes, and governance at multiple scales relevant to multi-(hazard-)risk management.
Acknowledging the number of international and intergovernmental efforts to harmonise and build common understanding of disaster risk related terminology, the *MYRIAD-EU Handbook* (Gill et al., 2022) adopted and built upon this existing work, using literature review, survey, and workshops to integrate different insights from across and beyond the MYRIAD-EU project. Our review of the literature identified that despite there being well-established definitions for a number of risk-related terms, many publications created their own, emphasising the need for a community-wide common understanding of
terminology (Gill et al., 2022). Following recommendation 3 of the UNDRR Technical Working Group on Hazard Definition Classification Review (UNDRR, 2020) (i.e. engaging with users and sectors for greater alignment and consistency



of hazard definitions), a participatory, co-developed approach was applied to the identification of terms and concepts of relevance to the project. Consortium members (researchers and sectoral representatives) were invited to submit terms and concepts for inclusion in the handbook, which were collectively discussed during a participatory session at the first General

Assembly and through subsequent online meetings. There are 140 terms in the glossary, 102 directly related to multi-(hazard-)risk management and an additional 38 due to their specific relevance to the project (see Gill et al., 2022). An early learning in the process was the acknowledgement that divergences in definitions are necessary (e.g. sector specific defined terms) and that the ambition should be for a common understanding rather than standardisation of definitions. This and other insights were shared at a 2-day international workshop in April 2022 (62 participants) held to gather perspectives from

experts from outside the MYRIAD-EU project.

The Handbook took a deeper dive into indicators for multi-hazard risk, leading to a collaboration with the MEDiate project to extend this into a wider review of multi-hazard indicators (White et al., 2025). The terminology was used by the Pilots to understand the different concepts within the project. It also brought a notable contribution to sectoral understanding of disaster risk terminology (e.g., the finance sector) and has been adopted by other European research projects as the basis for

their own terminology and concepts (e.g., Kennedy et al., 2023).

The MYRIAD-EU partnership spanned different academic disciplines, different sectors and different languages, meaning that building a common understanding was not without its challenges. The Handbook was delivered in English, and would have benefited from being translated into local languages to facilitate the engagement of decision-makers; for example, parts of the handbook have been translated in Norwegian in a project for the municipality of Bergen. Ideally the process would

have benefited from greater insight into the different disciplinary backgrounds and epistemological perspectives of the expertise across the partnership. Moreover, despite reaching consensus at the point of publishing the handbook (month 9 of the project), implementation of the terms and definitions only began in earnest through the subsequent technical work packages and application in the Pilots. Consequently, as the project progressed, additional terms emerged and some existing definitions in the handbook were contested when applied in practice. A key learning was therefore that building a common

understanding is an evolving process and the handbook needed to be 'living'. This is partly enabled by adding the glossary to the Disaster Risk Gateway wiki (https://disasterriskgateway.net/). The wiki is a project output created for the discovery and sharing of multi-(hazard-)risk quantitative and qualitative frameworks, models, tools and methods, and resources across the disaster risk community. The content in Disaster Risk Gateway can be regularly updated, fostering collaboration and encouraging contributions from the wider disaster risk management and climate change adaptation communities in building

a common understanding.

A key step in establishing a common baseline was reviewing existing multi-(hazard-)risk assessment and management approaches, building upon existing reviews and the knowledge and experience across the consortium. This process evidenced a predominance towards risk assessment rather than risk management approaches to multi-(hazard-)risk, which is demonstrated in the examples catalogued within the Disaster Risk Gateway. The target audience of Disaster Risk Gateway

includes researchers/academics; practitioners, policymakers and educators and students, and has facilitated



collaboration/conversations across Horizon Europe funded multi-(hazard-)risk projects on opportunities for co-development of information platforms. While scientific research continues to advance conceptual frameworks and promising methodologies, their practical application remains limited (Schlumberger et al., 2023). Many tools for identifying and managing hazard interactions are still in early development, and their usability for practitioners is often constrained by

technical complexity or lack of contextual relevance. Addressing this gap requires not only refining these tools but also embedding risk assessment and management processes within integrated, real-world decision-making contexts.

The development of strategic solutions to multi-(hazard-)risk challenges requires a good understanding of the decision context and processes that govern multi-(hazard-)risk. Our research therefore also investigated and reviewed policies, policy-making processes and governance for multi-hazard risk management in Europe (Schlumberger et al., 2023) using both

literature reviews and interviews with sectoral representatives. This work identified a need to increase awareness of inter-sectoral dependencies, and also provided several lessons and insights for research, policy, practice and co-production.

Multi-(hazard-)risk governance is a deeply interdisciplinary endeavour, shaped by complex interactions across hazards, sectors, and systems. Despite the central role of National Risk Assessments (NRAs) in disaster governance, they rarely reflect the systemic ambition of multi-(hazard-)risk policies. Emerging platforms and partnerships offer opportunities for

cross-sectoral collaboration, yet these remain fragmented and underutilised. Practitioners often struggle with siloed systems and limited guidance, making effective communication, through intuitive tools and tailored narratives, essential for fostering shared understanding and actionable strategies. As climate change intensifies risk interdependencies, some sectors are beginning to reconsider long-term governance structures.

## 2.2. Co-developing a framework for multi-(hazard-)risk assessment and management

Responding to the identified lack of a harmonised framework and guidelines for multi- (hazard-)risk assessment and management, we co-developed a framework for systemic, multi-(hazard-)risk assessment and management (Hochrainer-Stigler et al., 2023). A guiding principle was that it should overcome the classic approach where individual hazards/sectors are the point of departure. The framework was co-produced through an iterative process involving stakeholders from the Pilot regions and beyond, including stages of development, testing, feedback, and refinement. Prototype framework

development started with a thorough literature review of existing frameworks tackling hazards and related risks and a discussion among MYRIAD-EU experts. This prototype was then presented and discussed through a two-day international workshop in April 2022 (62 participants) with experts, practitioners and scientists (Hochrainer-Stigler et al., 2022). It quickly became clear that the framework needed to be flexible enough to operate on the spectrum from single to multi- and systemic risk analysis. In this way, it can accommodate different existing tools and methods that are already in use and can be tailored

to the specific context.

The framework was built on the concept of dependencies, which enabled the conceptualisation of single to multi-risks within a risk continuum. From a research perspective, it had the additional benefit that direct risks (e.g. due to the hazards





themselves) and indirect risks (e.g, follow-on consequences and cascading effects) could be further integrated to provide a comprehensive picture of impacts for different stakeholders.

Following the workshop, the prototype framework was revised into our initial six-step framework (Hochariner-Stigler et al., 2023), which is summarised in Fig. 1. Each step is accompanied by a 'guidance protocol', which is essentially a set of questions to be asked when working through a specific step of the framework.

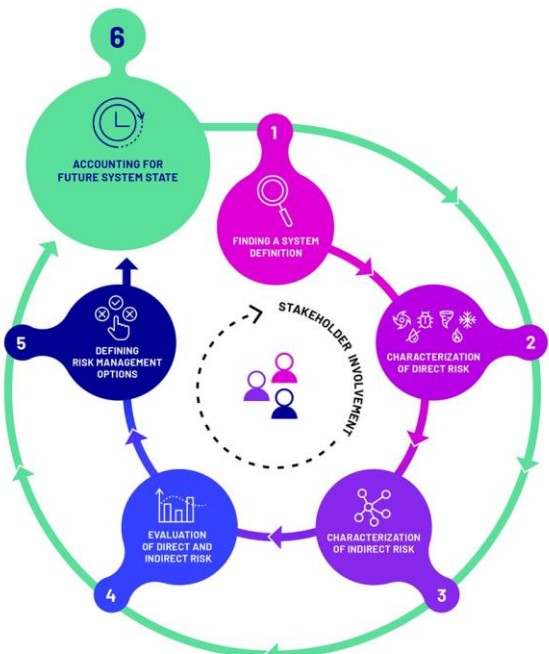

**Figure 1: The MYRIAD-EU six-step framework for individual, multi-, and systemic risk analysis and management (simplified**
**figure adapted from Hochrainer-Stigler et al., 2023, CC BY 4.0).**

This initial framework was then iteratively tested within the five Pilot studies. In practice, the framework was used to provide a "frame one can work with", using a system-of-systems approach based on boundary concepts. In this way, the different stakeholders involved could identify elements that belong to their "system" (e.g. political boundaries, business models or sectoral aspects), including resources and instruments in their repertoire to deal with current and emerging risks.

By doing this, it was also possible to concretely identify gaps as well as opportunities between different systems, such as governments, insurers or households, and the respective roles they could play in managing risk. The system boundaries concept was also useful with regards to policy-making processes. Policies are often related to specific goals that are designed for specific single hazards or risk situations and can be brought together through a step-by-step approach that is in its essence, a co-production process (Hochrainer-Stigler et al., 2023). Indeed, the framework itself provides a clear structure on

how single as well as multi-risks can be thought of simultaneously with increasing complexity, therefore making it possible



to develop multi-(hazard-)risk management strategies for current and future risks that was further developed within the Dynamic Adaptive Policy Pathways - Multi Risk (DAPP-MR) approach discussed in Section 2.5.

The main learnings from the implementation in the Pilots show that its structured, six-step design effectively supported users in navigating the complexity of multi-(hazard-)risk assessment. In MYRIAD-EU, the framework was applied at diverse geographical scales, from local to multinational, demonstrating its flexibility. Pilots found that the framework promoted systems thinking and enabled a more holistic understanding of interdependencies across hazards, sectors, and governance levels. Its flexible and non-prescriptive nature allowed for adaptation to diverse regional contexts, facilitating tailored approaches to risk analysis and stakeholder engagement. Importantly, the framework helped clarify DRM processes in the regions, supported the integration of direct and indirect risks, and fostered more meaningful dialogue with stakeholders by serving as a shared reference point throughout the process.

In order to make the framework more accessible, we have developed an online dashboard (https://dashboard.myriadproject.eu/), to help users to understand and navigate it. The dashboard contains the guidance protocols to carry out each of the six steps, provides examples of how the framework has been implemented in five Pilot regions, and examples of tools and methods that can be used to carry out the analysis within each step. The framework is now being used beyond the initial project. For example, the framework and/or parts of its steps are being used in EU-projects such as DIRECTED, CLIMAAX, and P2R.

## 2.3. Increasing understanding of dynamic feedbacks between hazard, exposure, and vulnerability

We developed a range of methods to increase our understanding of dynamic feedbacks between hazard, exposure, and vulnerability. These are brought together in the MYRIAD-EU Metadatabase for Dynamics of Risk Drivers (Stolte et al., 2025). This online metadatabase provides a structured catalogue of empirically grounded methods and datasets, developed both within and beyond the MYRIAD-EU project, to assess evolving risk patterns and feedbacks across spatial and temporal scales. We identify four key themes of methods developed within the MYRIAD-EU project: (1) multi-hazard dynamic footprints and susceptibility maps; (2) time-dependent intensity-damage functions; (3) vulnerability indicators; and (4) exposure as a function of time since a previous hazard. A synthesis of the key methods developed along each of these themes is provided below.

In terms of multi-hazard dynamic footprints and susceptibility maps, we show that unsupervised machine learning, specifically DBSCAN, can create spatiotemporal footprints of multi-hazard events/clusters and extract information on the single and multi-hazard event duration, intensity, extension, and seasonality under current and future conditions (Ferrario et al., 2025a). Furthermore, we developed a global framework for multi-hazard susceptibility mapping using deep learning, applying it to Japan (Tiggeloven et al., 2025b) and the Veneto region (Ferrario et al., 2025b). These susceptibility maps show spatial representations of the likelihood of specific hazards (such as landslides or floods) occurring in different geographic areas, trained on historical hazard-influencing factors such as topography, geology, hydrology, land use, and vegetation.



They can provide practitioners with spatial tools to identify hotspots over time where multiple hazards occur (Tiggeloven et al., 2025b). Building on this foundation, our review of AI applications across all four pillars of the Early Warning for All (EW4ALL) framework reveals significant opportunities to enhance the effectiveness of early warning systems, while also identifying critical gaps in knowledge, application, and policy that must be addressed for responsible implementation (Tiggeloven et al., 2025c).

In terms of time-dependent intensity-damage functions, we identified and analysed impacts from single-hazard and multi-hazard events in the global emergency events database EM-DAT to understand potential differences in impacts due to multi-hazard dynamics. The results indicate that whether or not multi-hazard impacts are different to the sum of single-hazard impacts depends on the type of hazard as well as on the type of impact (Jäger et al., 2025). A follow-up study, using machine learning and explainable AI techniques on compound flood events in Europe, showed that a higher number of co-occurring hazards significantly amplifies damages, particularly in vulnerable regions (De Ronco et al., 2025). Machine learning, neural networks, and explainable AI approaches can help to enable the operationalisation of these complex, non-linear relationships by capturing dynamic interactions between hazards and evolving impacts (Del Barco et al., 2024, 2025); however, we propose guiding questions for responsible AI before making such models operational (Tiggeloven et al., 2025c). Further, we demonstrate that risk components vary spatially and temporally, and that identical hazard intensities can produce different impacts depending on multi-hazard conditions (De Polt et al., 2025a), the timeframe that is being assessed (De Polt et al., 2023), or dynamic vulnerability changes (Schlumberger et al., 2025a). Developing such hazard(s)-impact relationships ensures more informed relationships to set warning criteria and input for risk assessments. More broadly, a better understanding of dynamic feedback of risk drivers allows practitioners to make more informed decisions and design more effective DRM measures (de Ruiter et al., 2020).

The need for more qualitative approaches was underscored by our stakeholder interviews across the five Pilots (van Maanen et al., 2025), which showed rich contextual evidence of dynamic changes in vulnerability and exposure, shaped by climate change, governance structures, and socio-economic shifts, which would be difficult to capture using conventional assessments. In this sense, disaster forensic studies (UNDRR, 2024), which examine systemic root causes rather than immediate impacts, can also be useful. We applied the paired-events disaster forensics method of Kreibich et al. (2017, 2022) to examine multi-(hazard-)risk dynamics across 57 multi-hazard events, bringing together over 150 experts globally, the results of which demonstrate how risk can be affected by sequences of events; an accompanying database will be published in a forthcoming paper (Šakić Trogrlić et al., 2025).

In terms of vulnerability indicators, we show that vulnerability to multi-(hazard-)risk is unequally distributed around the world, with socioeconomic conditions, such as human development index, political stability, and distance to healthcare, playing a critical role (Tiggeloven et al., 2025d). Stolte et al. (2024) designed *VulneraCity*, a database of 1,460 vulnerability drivers from over 400 papers. By analysing these drivers, we highlight the complexity of interactions among risk components in multi-(hazard-)risk settings, and identify six types of directional dynamics that challenge the assumption that



vulnerability can be adequately captured through static, generalisable indicators. In the context of flood risk research, we further investigated how different assessment methods are used to capture different types of dynamics, considering different temporal focus, dimensions of vulnerability, and sources of data used (Schlumberger et al., 2025a). Our stakeholder interviews across the Pilot regions reinforce the need for vulnerability assessment and management approaches that are

sensitive to local conditions and hazard-specific dynamics (van Maanen et al., 2025).

We highlight that a lack of recovery between events can erode a society's ability to prepare for, cope with, and recover from new disasters, potentially significantly prolonging recovery trajectories (Buijs et al., 2025). In terms of exposure as a function of time since a previous hazard, Buijs et al. (2023) use nighttime light satellite data to compare general trends in impacts and recovery for single and multi-hazard events on a global scale. This showcases the potential of Earth Observation

data as a valuable tool for tracking dynamic processes like recovery over time in multi-(hazard-)risk contexts, helping to identify where resilience is being impaired between events, which is something that should be further explored to support timely and data-driven interventions (Van Maanen et al., 2024).

## 2.4. Developing future scenarios of plausible multi-(hazard-)risk

As identified in Ward et al. (2022), many datasets, papers and tools exist for analysing current and future single hazards. In

MYRIAD-EU, we observed a lack of multi-hazard scenarios, methods to evaluate their direct and indirect risk, and user friendly software to evaluate such scenarios and risks. Within the project's efforts to establish a common set of multi-risk definitions (Section 2.1), we distinguish between current scenarios (based on historical or present-day data) and future scenarios (which also incorporate projected climate metrics) (Gill et al. 2022). These scenarios support understanding and communication of risk and can take various forms such as historical data, probabilistic indicators, or storylines.

While Ward et al. (2022) called for future scenarios of plausible multi-hazards, our early discussions with stakeholders in the Pilots revealed that the understanding of current multi-hazard events remains very limited, with no widely applicable event sets beyond case studies. To address this gap, we developed the MYRIAD-Hazard Event Set Algorithm (MYRIAD-HESA) (Claassen et al., 2023), which compiles single-hazard footprints into multi-hazard event sets based on their spatial and temporal overlap. Using eleven hazard types, we created the first global multi-hazard event set database, enabling the

identification of hotspots, frequent hazard pairs such as heatwave-drought and cyclone-flood, and more complex groups like multiple earthquakes followed by tsunamis and landslides.

MYRIAD-HESA shows us what has happened in the past, however, methods are also required to evaluate what could happen in the future. Within MYRIAD-EU this is achieved through the use of stochastic data and storylines. Stochastic data are plausible events based on the statistical properties of historic observations. Within MYRIAD-EU, we developed a

stochastic earthquake set for Europe (Schaefer et al., 2022). We also developed MYRIAD-SIM, a weather generator that simulates extreme variables while preserving their dependencies using copulas (Claassen et al., 2025a), and VineCopulas (Claassen et al., 2025b), an open-source Python package for vine copula modelling. MYRIAD-SIM was used to evaluate the





likelihood of high impact multi-hazard extreme weather events, such as the occurrence of multiple storms in close succession. The results show that while some multi-hazard events are rare, they may reach higher intensities than those
observed historically, meaning they may pose greater risks than suggested by past records alone. To complement such probabilistic approaches, multi-(hazard-)risk storylines offer an alternative by focusing on plausibility rather than probability. MYRIAD-EU has developed an approach to storylines that explores cause-and-effect relationships in multi-(hazard-)risk systems (Crummy et al., 2025). Drawing on past events for context, MYRIAD-EU explored the application of storylines to support multi-hazard decision-making through co-creation and iterative exploration (Crummy et al., 2025;).
This approach supports understanding of complex events and the development of forward-looking decision pathways in the Pilot regions (Sections 2.4 and 2.5). The development of storylines supports a holistic examination of cascading effects, critical thresholds and emerging risks (Cocuccioni et al., 2025).

Within MYRIAD-EU, we also studied indirect impacts, such as disrupted supply chains that cause economic losses for businesses outside the hazard zone, and showed that indirect impacts strongly affect the overall impacts of hazard events by
evaluating common sets of hazard events across macro-economic models (IIASA-ABM (Poledna et al., 2023), GRACE (Aaheim et al., 2018), MRIA (Koks and Thissen, 2016). Through the examination of specific scenarios like the Danube Region's earthquake and flood events, and the Scandinavia region's heat, drought, and wildfire events, we demonstrate that indirect impacts can influence regional and sectoral economies in ways not immediately apparent from direct damage assessments alone. Our results reveal that regions not directly affected by a hazard event can both benefit from shifting
demand and increased economic activity due to reconstruction efforts as well as suffer from missing inputs produced in affected regions (Ducros et al., 2024). However, the magnitude of the both indirect and total impacts varies strongly across modelling approaches.

To complement the varying modelling approaches and make multi-(hazard-)risk scenarios more accessible to a broader audience, a user-friendly online software has been developed (https://www.myriad-multirisk.eu/). The MYRIAD-EU
Software combines a range of different data into one place, such as vulnerability curves for different hazards, and both current and future hazard exposure data in the exposure at risk explorer. A key learning from the development of multi-(hazard-)risk scenarios was that different sectors require different risk metrics. Therefore, the software provides multiple exposure indicators such as GDP, population, and tourism expenditure, allowing sector-specific assessment of multi-(hazard-)risks. The software is open-source, available online for anyone to use, and will continue to be maintained, tested, and further
developed beyond the end of the project.

## 2.5. Assessing the effectiveness of DRM measures across hazards, sectors, and time horizons

Our research agenda stressed the need to account for uncertainty in future changes and cross-sector, multi-hazard interactions in DRM to leverage synergies and avoid trade-offs. It observed a lack of guidance for contexts involving multiple hazards, sectors, and timescales, and identified pathways thinking as a promising foundation for advancing multi-(hazard-)risk policy



analysis. As one approach, in MYRIAD-EU we extended the Dynamic Adaptive Policy Pathways (DAPP) into a multi-risk framework (DAPP-MR; Schlumberger et al., 2022).

DAPP-MR guides the evaluation of adaptation pathways under multi-(hazard-)risk conditions. It systematically considers three themes critical for DRM across multiple hazards, sectors and time: (1) interactions between multiple hazards; (2) sectoral dynamics and interdependencies; and (3) synergies and trade-offs of DRM options across sectors and scales. Building on the MYRIAD-EU framework (Section 2.2), DAPP-MR applies systems thinking and introduces complexity stepwise: pathways are first developed per hazard and sector, then multi-hazard effects within sectors are added, before integrating all hazards, sectors, and time dimensions (Fig. 2). A key learning is that while complexity is essential, the process must remain tractable. Iterative integration provides interim results that both inform policy and simplify analysis under more complex dynamics.

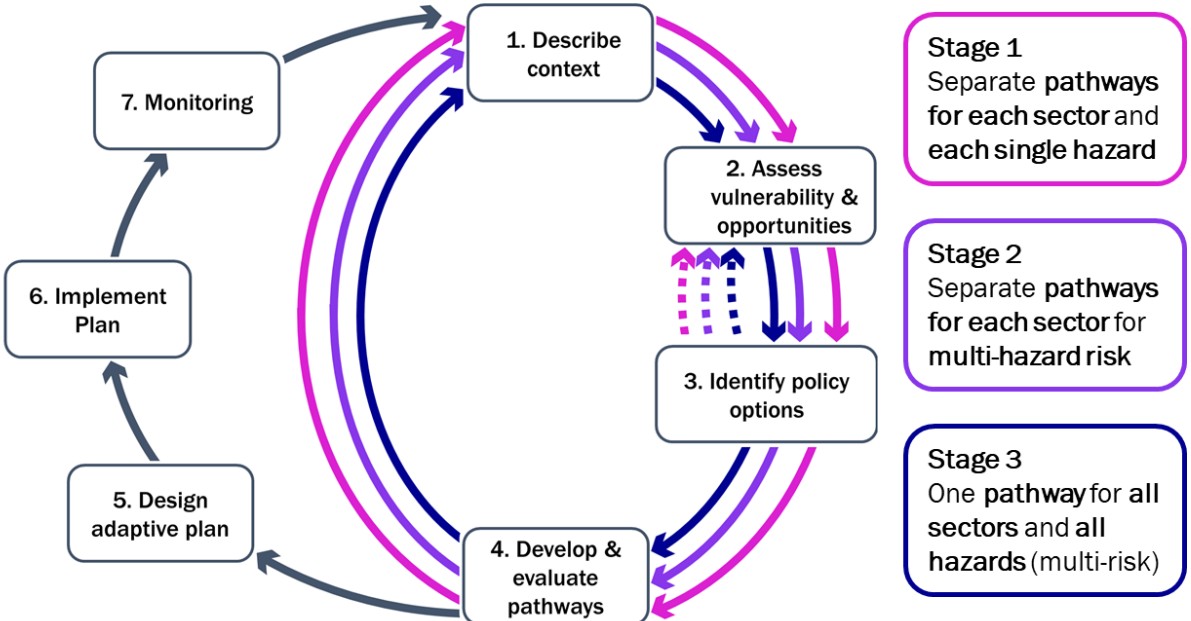

**Figure 2: The DAPP-MR framework to guide the development and evaluation of pathways with increasing complexity from multi-hazard and multi-sector interactions through time. Source: copied from Schlumberger et al. (2022), CC BY 4.0**

*Quantitative evaluation of DRM effectiveness*

We applied DAPP-MR in a synthetic multi-risk case study (Schlumberger et al., 2024) to evaluate DRM effectiveness quantitatively. The results reveal the difficulty of assessing effectiveness of DRM in multi-(hazard-)risk contexts. Some DRM measures gained new relevance or were conceptualised differently: for example, nature-based flood measures such as extended floodplains also improved groundwater recharge under drought conditions. Interactions between pathways, their timing, and the presence of other DRM measures strongly shaped the exposure and vulnerability and thus ultimately the risk





outcomes. Managing risks in isolation risks overestimating measure and pathway performance, but the staged approach
highlights pathways that are robust under increasing complexity.

Effective visualisation was found to be critical. Decision support in complex systems requires both data and judgment about
which analyses and data matter most (Hadjimichael et al., 2024). To this end, we developed an exploratory pathways
dashboard, which enables stakeholders to interactively explore the quantitative data relevant to assess the effectiveness of
DRM across sectors, hazards, and time (Schlumberger et al., 2025b).

Beyond the pathways approach, additional approaches have been used to evaluate the effectiveness of DRM measures across
hazards, sectors and time quantitatively. For example, in the work mentioned in Section 2.3 by De Polt et al. (2023), we
developed qualitative impact functions between heat-related hazard intensities and impacts (e.g., fatalities), which show how
actions taken by individuals can lower their vulnerability.

*Qualitative evaluation of DRM effectiveness*

DAPP-MR was also applied qualitatively in four Pilot studies (Gottardo et al. 2025, Schlumberger et al. 2025c). In the North
Sea Pilot, a structured interaction matrix assessed sectoral impacts of measures, identifying synergies and trade-offs in their
effectiveness across hazards and sectors. This supported evaluation of cross-sectoral versus sectoral optimisation. A similar
approach in the Danube Region Pilot focused on timing and coordination between agriculture and shipping, with
stakeholders co-developing sectoral pathways simultaneously to discuss synergies and trade-offs and highlighting
governance and land-use challenges.

The Scandinavian Pilot adopted a systems-level perspective, facilitating broad, accessible discussions of cross-sector DRM
without explicit measure-level analysis. The Canary Islands Pilot used quantitative modelling to link tourism development
and water availability, showing how sectoral strategies compounded climate pressures. Visualisation tools enabled
exploration of trade-offs and the feasibility of water-saving measures under interaction scenarios, including "what-if" cases
where tourism and agriculture were simultaneously strained by limited water supplies during heatwaves.

Across Pilots, DAPP-MR fostered integrated, cross-sectoral thinking and stakeholder dialogue. In the Canary Islands, it
helped to structure discussions between stakeholders from the tourism and agriculture sectors, highlighting governance gaps
between water planning and risk reduction. In the Danube Region, it helped to improve awareness of interdependencies in a
transboundary context. In the North Sea, it helped to initiate new stakeholder networks in a region with limited DRM
awareness. Overall, the framework facilitated shared understanding of multi-(hazard-)risk systems and supported solution-
oriented dialogues on synergies, trade-offs, and long-term planning. All Pilots demonstrated the value of scenario-based
thinking in the evaluation of DRM effectiveness: beyond climate scenarios, they incorporated dimensions such as social
perceptions (Scandinavia), spatial planning (North Sea, Scandinavia), external decision-making (Canary Islands), and socio-
economic uncertainty (Danube Region) to identify DRM options that were either feasible or necessary.





A consistent finding was that building a shared system understanding, including sectors, vulnerabilities, and short- and long-term needs, required significant effort but eased integration of additional complexities (Schlumberger et al., 2025c). In some cases, multi-(hazard-)risk storylines of past events facilitated this process, improving engagement and integration of uncertainty (Crummy et al., 2025).

Qualitative evaluation also highlighted challenges. Tracking all interaction effects and trade-offs was difficult without 370 quantification. Still, qualitative analysis often revealed additional measures with strong cross-sector or multi-(hazard-)risk synergies and allowed for a richer characterisation of these interactions. Pilots therefore adopted semi-quantitative criteria to structure DRM evaluation.

Beyond DAPP-MR, additional approaches have been used to evaluate the effectiveness of DRM measures across hazards, sectors and time. multi-(hazard-)risk storylines (Crummy et al., 2025) were applied in different Pilots primarily with a focus 375 on understanding past events and the effectiveness of DRM measures. Storylines facilitated Pilots to engage with stakeholders on multi-(hazard-)risk interactions, facilitating discussions on plausible future events and multi-(hazard-)risk decision-making. There is large potential for storylines to be used in practice for supporting development of DRM pathways or the evaluation of DRM effectiveness, which should be explored in future multi-(hazard-)risk projects. In the Veneto Pilot, the use of storylines to inform multi-(hazard-)risk and cross-sectoral pathways was tested by combining climate storylines 380 with the Peer Review Assessment Framework (PRAF), developed under the Union Civil Protection Mechanism (Casartelli et al., 2025). The PRAF, which provides a flexible and comprehensive structure spanning the entire DRM cycle, as well as risk governance, risk assessment, and DRM planning, facilitated the identification of multi-(hazard-)risk pathways, in alignment with existing regional policies and practices. The integration of co-created storylines, projecting a past storm event (the Vaia storm of 2018) 56 years into the future (Vaia 2074) enabled the identification of detailed cross-sectoral pathways by 385 adopting an holistic, integrated, and participatory approach.

Beyond the pathways approach, additional approaches have been used to evaluate the effectiveness of DRM measures across hazards, sectors and time quantitatively. For example, the disaster forensic study by Šakić Trogrlić et al. (2025) shows how DRM measures taken between paired-events provide evidence of social learning and missed opportunities to learn from past events. Moreover, the work carried out to develop the VulneraCity database (Stolte et al., 2024) and take stock of dynamic 390 vulnerability assessment methods (Schlumberger et al., 2025a) provides qualitative information on how DRM actions can influence the vulnerability in urban or flood contexts.

## 2.6. Testing of approaches in in-depth case studies on multi- (hazard-)risk assessment and management

Our five Pilots tested and refined a diverse set of innovative tools, methods, and approaches that address multi-(hazard-)risk and multi-sector challenges in DRM in varied geographic and socio-economic contexts, including the aforementioned 395 MYRIAD-EU framework, the DAPP-MR approach, and the other tools and methods described in the preceding sections (Gottardo et al., 2025). Through this process, all five Pilots came across the following challenges: (1) limited cooperation



between institutions (government authorities, public agencies, private entities) within and across countries and sectors; (2) limited stakeholder awareness and engagement, mainly due to capacity constraints; (3) limited data availability and quality, along with limited understanding of climate and non-climate risk components and their interactions; and (4) complexity of the system (multi-(hazard-)risk, dynamic risk, multi-sector, multi-scale).

Despite these challenges, the Pilots built a solid stakeholder network and co-developed multi-(hazard-)risk pathways relevant for local and regional as well as national and international actors. Based on this experience, it is possible to draw a set of key learnings for researchers and experts:

- *Stakeholders need to be involved in the co-development process from the early stages of any research project.* A relevant group of stakeholders includes government and public agencies, research and academia, as well as representatives of the private sector and the civil society (specific business operators, municipalities and citizens). Stakeholders should also be selected with knowledge and/or responsibilities across hazards, sectors, scales and countries. Their involvement from the early stages ensures that research results meet their needs and are actionable. Diversifying the engagement fosters participation and exchange (through, for example, workshops, webinars, interviews).
- *An effective communication is key to ensure alignment between project objectives and stakeholder expectations.* It is important that researchers and experts raise awareness about the current and future risks in the region and honestly communicate what can be done and what cannot be done with the available methods and tools as well as the associated uncertainties. Results need to be understandable, usable, and tailored to the specific needs of the involved stakeholders.
- *A 'one-size-fits-all' approach is not suitable and should not be applied at any level (transnational, national, local, sectoral).* Every region in Europe faces different challenges, which depend on factors such as scale, relevant hazards, relevant sectors, governance landscape, financial capacity, therefore a tailored range of solutions in terms of methods and tools should be offered. In MYRIAD-EU, using combined quantitative and qualitative methods has proven to be crucial in disentangling complexities, filling in any existing gaps in knowledge, and communicating the results to the involved stakeholders.
- *Multiple data sources (e.g., from in situ to Earth Observation-based) should be considered and integrated in applications to increase data availability and quality.*

We have also drawn key learnings for government and public agencies, which, if implemented, could facilitate the replicability and upscaling of MYRIAD-EU research outcomes:

- *Apply multi-criteria decision-making in policy design and DRM or adaptation planning.* Governments and public agencies should move beyond relying on single indicators, such as effectiveness or cost-efficiency and incorporate additional decision-making criteria, such as long-term resilience, sustainability, spatial equity, and risk tolerance, when evaluating policy options and DRM or adaptation pathways. This ensures more robust, transparent, and context-sensitive choices that reflect diverse stakeholder priorities and system trade-offs.
- *Support the mapping of institutional roles and governance responsibilities at the onset of any collaborative initiative.* Governments and public agencies should support partnerships in mapping existing institutional frameworks, decision-making levels, and governance arrangements at the relevant supra-national, national, regional, or local level. This helps to clarify mandates, prevent overlaps or gaps, and align new actions with existing structures, particularly in complex or multi-jurisdictional contexts.





- *Create or strengthen existing cross-sector, multi-level coordination mechanisms*. Permanent coordination bodies initiated by public authorities to facilitate alignment across sectors (e.g. water, agriculture, tourism, energy, forestry) and governance levels (municipal, regional, national) can support a more integrated approach on future DRM and climate change adaptation. These may include interdepartmental working groups, joint planning platforms, or shared data systems to enhance coherence in climate adaptation and risk governance strategies.
- *Recognise and engage informal and non-state stakeholders in governance processes*. Formally, recognise roles of non-state actors, such as private water associations, as key contributors to an increasingly complex risk governance landscape. Develop inclusive engagement strategies that bring these stakeholders into planning and decision-making processes from the outset, thereby increasing policy legitimacy, ownership, and effectiveness.
- *Align policy objectives across sectors to enhance coherence and avoid contradictions*. Develop, review, and revise sectoral policies considering interdependencies between sectors (or policy areas) and to ensure consistency between climate adaptation, disaster risk reduction, and sustainable development goals. Use scenario-based planning approaches to identify potential policy trade-offs and facilitate inter-ministerial coordination to promote coherent, system-wide resilience strategies (e.g., via a designated network of senior level servants attached to each sector or policy area).

## 2.7. Additional key findings

### 2.7.1. Knowledge co-production

The co-production process in MYRIAD-EU was both a methodological foundation and a practical challenge, shaping the development and implementation of multi-(hazard-)risk approaches across the project. A more detailed analysis of the co-production methodology and outcomes will be presented in a separate manuscript (building on Ciurean, 2025), this section synthesises key learnings relevant to the broader MYRIAD-EU research agenda. Figure 3 illustrates the seven key elements of the knowledge co-production process in MYRIAD-EU.

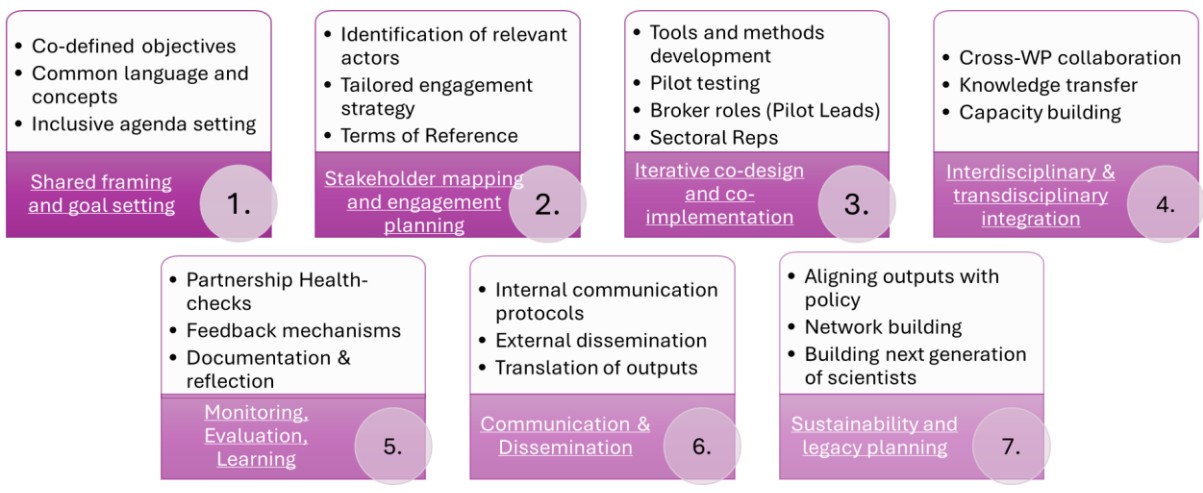

**Figure 3: Seven key elements of the co-development process in MYRIAD-EU. British Geological Survey © UKRI 2025**



Co-production was central to MYRIAD-EU's work packages and Pilots, enabling the integration of scientific and stakeholder knowledge in the development of frameworks, tools, and decision-support approaches. Before the co-development of any of these, MYRIAD-EU initiated the co-production process within the consortium with the development of the Handbook (Gill et al., 2022). This early effort underscored the challenge of establishing a shared understanding of key concepts and terminology across disciplines early on, while also recognising and integrating the diverse knowledge and

perspectives within the partnership.

Co-production was then key to developing the frameworks, tools, and decision-support approaches. For instance, in the creation and iterative updating of the MYRIAD-EU framework, co-production enabled a deeper understanding of the complex hazard interrelationships among stakeholders. This was important for defining system boundaries and recognising how dependencies between hazards and elements at risk could influence components both within and beyond the system.

Similarly, the co-production process underpinned the development of the multi-(hazard-)risk software, underscoring the importance of early engagement with end-users during scenario design. By integrating stakeholder input, particularly from the finance and tourism sectors, the scenarios were more effectively contextualised and tailored to local decision-making needs.

The co-production approach also played a key role in fostering knowledge exchange and in shaping both bottom-up and top-

475 down strategies for multi-(hazard-)risk assessment and management. Specifically, it ensured that the diverse insights, resources, and instruments available across different sub-systems were acknowledged and incorporated into an integrated strategy, i.e., one that enhances synergies and reduces asynergies.

Pilots showed that early and continuous stakeholder involvement was essential for tailoring outputs to local needs. Co-production helped translate complex scientific concepts into actionable insights, such as through the use of storylines,

dashboards, and sector-specific scenarios. The process fostered mutual learning and built trust, but also required significant facilitation effort and time investment, particularly from Pilot Leads who often acted as knowledge brokers, translating between technical teams and stakeholders. While this approach was largely successful, it underscored the importance of capacity-building and the need for more direct interaction between tool developers and end-users, which is often difficult to achieve in large-scale complex projects.

The co-production process supported cross-sectoral dialogue and helped identify governance gaps, particularly in transboundary and multi-level contexts; it also contributed to mapping the knowledge on multi-risk governance and decision-making processes early in the project. Stakeholders valued the opportunity to shape research outputs and saw potential for uptake in policy processes, especially where tools were co-developed with local actors. However, feedback also pointed to the need for clearer pathways to impact, including targeted dissemination to decision-makers and practical

examples of tool application. Future projects should consider integrating co-production strategies from the proposal stage, incorporating clear mechanisms to ensure sustainable exploitation and lasting impact beyond the project's lifetime.



MYRIAD-EU's experience reinforces that co-production is not a one-size-fits-all process. It evolved differently across Pilots, shaped by local contexts, stakeholder networks, and thematic priorities. Key enabling factors included a culture of openness, iterative feedback loops, and the use of participatory formats such as focus groups and workshops that were followed by an evaluation of the engagement. Challenges included managing expectations, navigating disciplinary boundaries, and sustaining stakeholder engagement over time. The project's evaluation (externally, via stakeholder surveys, and internally, via partnership health-check forms) showed generally high satisfaction with the interaction and collaboration processes, but also identified areas for improvement, as indicated above. Overall, MYRIAD-EU demonstrates that knowledge co-production strengthens the relevance, usability, and legitimacy of multi-(hazard-)risk research. However, it requires dedicated resources, skilled facilitation, and strong commitment to shared learning.

### 2.7.2. Early Career Researcher Empowerment

Early Career Researchers (ECRs) were key contributors to the advances and research outcomes of the MYRIAD-EU project. From the proposal phase, there was a strong commitment to providing means of empowerment of ECRs directly within the project's structure. This was especially important since approximately 30% of the entire consortium consisted of ECRs. One of the main structural components was the implementation of the Early Career Researcher Board (ECRB), which operated in terms of 2 consecutive years, enabling a number of ECRs to participate throughout the project's lifetime. The board had several tasks, including the organisation of ECR-centred formal and informal gatherings and ensuring the ECRs in the project were informed about any information relevant to them. A core component of the ECRB was the elected Early Career Representative, who was a full member of the project's Management Team (MT), providing an opportunity to directly relate ideas, updates, and concerns from the ECRs within the project.

We observed that the broad network formed within the large MYRIAD-EU consortium supports and advances inter- and intra-disciplinary collaborations, as well as connections between different sectors. ECRs were able to benefit from this network through repeated interactions with, and mentorship from, senior colleagues who spanned these breadths. Access to these broad professional networks served as a valuable resource for ECRs within the project, contributing to their own professional development and the project's research goals.

The experiences of MYRIAD-EU ECRs, as well as senior members of the consortium, provided input for a broader analysis that identified three key enabling factors and examples of empowerment of ECRs (Schlumberger et al., 2025d). These activities fell into three main categories: (1) structural involvement in project management; (2) organisation of events for ECRs - together with ECRs; and (3) forming networks beyond their own work. The investigation also highlighted three essential enabling factors that support an empowering and encouraging environment: (1) direct influences and advisory support; (2) agency of ECRs to enable action; and (3) external factors in the project context.



Both internal and external (e.g., external advisory board and project review board) feedback confirmed that ECR engagement and empowerment to be a key strength of the project. MYRIAD-EUs experience demonstrates that through integrating ECR empowerment into the structure, activities, and networks results in lasting benefits for the ECRs and the projects themselves.

### 2.7.3. Policy and practice implications

The key learnings described in this paper help to address the urgent call of the Sendai Framework MTR to "*...adopt multi-hazard risk reduction approaches that address all risks at source before they manifest as shocks or disasters...*" (UNDRR, 2023, p. 23). A specific contribution to global scale policy has been through the contribution to the updated HIPS (UNDRR & ISC, 2025; Gill et al., 2025). At EU level, tools developed in MYRIAD-EU to disentangle the contribution of different hazards to overall impact (Jäger et al., 2025) and MYRIAD-HESA (Claassen et al., 2023) are being used in several collaborative studies between the consortium and the EU Joint Research Centre, for example to explore vulnerability dynamics associated with compound flood impacts across European regions (Ronco et al., 2025). The MYRIAD-EU framework was identified as a useful approach for assessing risk throughout Europe in the EU Climate Risk Assessment (EUCRA) report (EEA, 2024). Through the Pilots work, policy impacts have been made at regional to local levels. For example, parts of the MYRIAD-EU framework are now being applied in spin off studies together with municipalities; the Veneto Pilot contributed to the Regional Strategy for Climate Change Adaptation (Veneto region 2024); the Scandinavian Pilot contributed to the risk assessment of the municipality of Bergen; and the Danube Region Pilot is contributing to the Preliminary Flood Risk Assessment in the Danube River Basin report prepared by the International Commission for the Protection of the Danube River (ICPDR), especially its chapter on Supporting Transboundary Activities. This report will form a crucial component of the updated Danube Flood Risk Management Plan (DFMP) scheduled for release in 2027. Finally, through close collaboration with our sectoral representatives, we have developed sectoral briefs that provide stakeholders within these sectors information on how the knowledge developed can be used within their sector (ecosystems & forestry, Cordier and Appulo, 2025; energy, Bulder et al., 2025; finance, Champion and Madappatt, 2025; food & agriculture, Campillo and Gonzalo, 2025; infrastructure & transport, Adesiyun, 2025; tourism, Khazai et al., 2025).

### 3. Outlook

Progress on understanding, assessing, and managing multi-(hazard-)risk has developed rapidly over the last five years. Within Europe, this is for a large part due to the combined efforts of the projects mentioned in the introduction, as well numerous projects not mentioned here. Through these projects, a flourishing community of researchers and practitioners has evolved. This is evidenced for example by the rapid growth of the multi-hazards sub-division of the European Geosciences Union (EGU). Launched in 2019, this sub-division hosted five sessions at the EGU General Assembly 2025, making it one of the larger Natural Hazards sub-divisions in terms of sessions. Moreover, the topic is central to many of the Working Groups of the Knowledge Action Network on Emergent Risks and Extreme Events (Risk-KAN). In a collaborative effort





between RISK-KAN, NatRiskChange, and MYRIAD-EU, the *3rd International Conference on Natural Hazards and Risks in a Changing World - Addressing Compound and Multi-Hazard Risk* (Tiggeloven et al., 2025a) attracted ~500 abstract
submissions from science and practice, highlighting the rapid growth of this field.

This is an encouraging sign for the future, as the topic continues to become even more important in policy and practice. Nevertheless, it remains a relatively young and rapidly developing research field. As such, there are still many avenues for continued scientific development. In this section, we provide a brief perspective on some of these potential avenues, building from our experiences over the course of the MYRIAD-EU project.

An essential avenue is to **continue the mainstreaming and mutual understanding of concepts and definitions**. Over the last years, the concept of "multi-hazards" has begun to filter through to mainstream media reports of several hazards (e.g., BBC, 2024). This is an important development, and ensuring continued attention will require clear concepts and definitions. Building a common understanding across different sectors also requires that terminology can and is effectively translated into other languages, as was demonstrated in Norway. It also requires collective effort from the research community to adopt
established terminology or clearly explain why alternative definitions are required, rather than simply defining terms afresh in publications (see Gill et al., 2022). Advancing the contributions and reach of Disaster Risk Gateway would help to amplify terminology and maintain a baseline of understanding across the disaster risk community. With advances in AI/LLM since the wiki was developed, there are opportunities to explore new approaches to enhancing the discovery of information contained within Disaster Risk Gateway. To ensure synergy and mutual understanding between different projects and
activities, project clusters can be beneficial. For example, by creating an informal cluster of projects around multi-(hazard-)risk, which involved annual meetings at the EGU General Assemblies, several cross-project activities arose during the last couple of years. These include sharing news on each other's newsletters, writing of joint papers, the organisation of a joint ECR-led summer school (the DRM Academy; Gargiulo, 2025), and co-organisation of training sessions at the EGU General Assembly (De Polt et al., 2025b). While this clustering occurred organically, due to the shared ambitions of the various
project management teams, more formal clusters can help to reduce this dependency and ensure longer term clustering even when old or new projects come and go. An example is the Societal Resilience Cluster of Projects, part of the Community for European Research and Innovation for Security (CERIS), takes this role. Through the SRC, a series of cross-project webinars has been developed, policy briefs have been (and are being) developed that synthesise policy recommendations across projects, and a stakeholder hub has been established.

We need to **continue developing a strong evidence base of how multi-hazard-risk both shapes, and is shaped by, risk dynamics over space and time**. Our Metadatabase for Dynamics of Risk Drivers (Stolte et al., 2025) is a first step into the direction of a structured catalogue of empirically grounded methods and data to assess evolving risk patterns and feedback across spatial and temporal scales. We invite the scientific community to actively contribute to the database, in order to share data, methods, and knowledge on this topic. A key direction involves advancing mixed-method approaches to assess



dynamics of exposure and vulnerability that integrate quantitative modelling, qualitative stakeholder engagement, and Earth
     Observation data. Other opportunities include the expansion of disaster forensics studies, and the development of AI- and
     EO-driven tools to analyse complex multi-hazard-risk relationships and nonlinear dynamics (e.g. Tiggeloven et al., 2025c).
     These innovations can help overcome the limitations of current static or linear risk models by embracing risk as an emergent,
     evolving, process shaped by interactions between environmental hazards, social systems, and feedback loops. A richer,
context-specific evidence base is also required to demonstrate the value of multi-(hazard-)risk DRM thinking and planning.
     For example, historical re-analysis of past events using DAPP-MR or similar frameworks could highlight how different
     decisions might have improved outcomes - especially when interactions between multiple hazards are involved. Similarly,
     the storyline approach proved effective in engaging stakeholders with systemic risk thinking, serving as a successful
     conversation starter and entry activity, and offers potential for further development in the multi-(hazard-)risk setting. Recent
years have also seen an expansion of the concept of multi-(hazard-)risk by including disease outbreaks and health impacts as
     these have a strong link to risk dynamics of space and time (Sairam and de Ruiter 2025; UNDRR 2022).

     Despite progress over the last five years, there is still a large need for **further developing methods for providing both
     current and future multi-(hazard-)risk scenarios**. Within MYRIAD-EU, different tools and datasets were introduced,
     such as MYRIAD-HESA, MYRIAD-SIM, susceptibility maps, and the multi-(hazard-)risk software. Future research can
build on these tools by expanding the breadth of the multi-hazard scenarios, for example, extending the simulated data in
     MYRIAD-SIM and the software with more stochastic hazards, such as landslides, volcanic eruptions, and tsunamis. This
     would allow for a broader exploration of hazard interrelations. Additionally, these tools could be further expanded through
     the inclusion of climate projections, as we showed in the Veneto Pilot for future scenarios of multi-hazard susceptibility.
     Another key finding within MYRIAD-EU was the need for a larger scale macro-economic intercomparison project to help
unpack the uncertainties associated with current modelling practices related to indirect risks, offering insights into the
     variability of outcomes and their implications for policy decisions. By integrating an array of models, more reliable
     ensemble estimates of indirect impacts could be obtained, enhancing confidence in projections used for climate risk
     assessments and policy planning. This collaboration would refine the quantification of indirect impacts and deepen
     understanding of their systemic nature across socio-economic contexts and geographic scales. Within MYRIAD-EU an
initial workshop was organised to kick off this large scale macro-economic model intercomparison project. During the
     workshop the initial set of experiments for the macro-economic model intercomparison (macroMIP) was agreed upon,
     refined into a formal experiment protocol in online follow up meetings. Future experiments are intended to incorporate
     expected forcings under climate change based on output from impact models organised within the Inter-Sectoral Impact
     Model Intercomparison Project (ISIMIP).

Effective DRM evaluation in multi-risk settings depends on **the availability of appropriate, solutions-oriented, usable
     tools**. To inform the assessment of DRM effectiveness qualitatively or quantitatively, relevant tools need to strike a balance



between completeness and accuracy on the one hand, and accessibility and complementarity on the other hand. MYRIAD-EU brought forward a suite of powerful methods and tools for the assessment of aspects of multi-risk that are relevant to be understood for DRM decision-making. They were developed and tested in parallel to the work in the Pilots to develop and

evaluate forward-looking DRM pathways and as such there was limited space and time for full alignment. Future efforts could focus on their integration and adding interoperable tools that can address different dimensions of multi-risk complexity relevant for DRM effectiveness. This includes improving usability and accessibility, and also ensuring that outputs of different products are directly usable within an operationalised framework for evaluating DRM options in multi-(hazard)risk settings.

As the field of multi-(hazard-)risk begins to mature, it is important that we **more explicitly include equity issues and equitable disaster risk reduction and adaptation**, given that vulnerable and marginalised groups are disproportionately affected by the compounding impacts of disasters (Tiggeloven et al., 2025d). Within MYRIAD-EU, we carried out a high-level assessment of the exposure of sexual and gender minorities to flood risk (Mortensen et al., 2025), and stressed the need to explicitly include all minorities in risk assessment and management. Haer and de Ruiter (2024) discuss that equitable

adaptation aims to transform how risk is managed. This can be achieved through three aspects: (1) ensuring burdens and benefits of implementing adaptation measures are distributed fairly to those most in need (distributive equitability); (2) make sure that vulnerable groups and minorities are included in and have control over the adaptation decision-making processes (procedural equitability); and (3) addressing the underlying socio-economic inequalities that are the root causes of vulnerability (systemic equitability). By focusing on these aspects, adaptation efforts can prevent multi-hazard events from

pushing already vulnerable populations into a vulnerability trap and can create more resilient, socially just outcomes. To support this work, future research requires an even greater emphasis on the social dimensions of risk and vulnerability, and should develop new metrics that measure success in terms of community agency and social cohesion rather than just economic losses.

It is essential that we **continue extensively testing and coproducing multi-(hazard-)risk knowledge in in-depth case**

**studies** to ensure the transferability and scalability beyond isolated case studies or Pilot regions (van Maanen et al., 2025). Firstly, this means continuing the work started through MYRIAD-EU (and related projects) in the existing Pilot regions. Much time and effort has been invested in developing strong relationships and trust with, and among, stakeholders in these regions. As with many projects, being able to push this forward would allow for taking implementation of this knowledge to a next level. Efforts have been made in continuing collaboration with the established stakeholder networks to further

promote MYRIAD-EU results within their organisation and inform ongoing and future policy initiatives (e.g. regional climate adaptation plans, national risk assessment guidelines). However, funding mechanisms that allow for (and provide financial resources for) continuation of successful co-production environments would be instrumental in catalysing this but are currently very lacking. Secondly, the approaches tested in the Pilots are considered to be transferable and scalable to





other regions in the EU and beyond, provided that they are adapted/tailored to the local contexts. Therefore, the Pilots now

can serve as lighthouses for other European regions facing similar challenges and aiming to transition towards integrated, systemic multi-risk assessment and management, such as the EUCRA hotspots. Expanding this work to other regions both within and outside Europe is a key consideration, as transferring knowledge and methodologies to areas with limited or lower-quality data poses significant challenges. Variations in data availability and consistency across regions can affect the applicability and reliability of the tools. Some of our methods have, however, already been applied to regions beyond the EU

border (e.g., susceptibility maps for Japan) Pilot leads have already initiated projects aimed to upgrade their results or to apply their approaches to other regions or scales (e.g. ECOAMARE, AQUAMAN, EO4MULTIHAZARDS, and collaborations with municipalities to account for context-specific aspects linked to climate adaptation) and will put effort in finding new opportunities for transferability and upscaling in future Horizon Europe calls. In particular, we see an important avenue to be testing the approaches within Global Majority settings.

An overarching and promising avenue is to more explicitly examine how the multi-(hazard-)risk tools, methods, and knowledge developed can **support the development of Multi-Hazard Early Warning Systems (MHEWS)**. MHEWS are increasingly recognised as a global priority, as reflected in the Early Warning for All (EW4ALL) initiative launched by the UN Secretary-General in 2022, which aims to achieve universal coverage by 2027. Yet, recent analysis by Budimir et al. (2025) highlights that much of the current discourse and practice still emphasises "multiple single-hazards" rather than

addressing the interrelations between hazards and the dynamics of other risk components. Advances within MYRIAD-EU and related projects demonstrate clear potential to move towards truly multi-hazard-oriented systems. For example, tools such as MYRIAD-HESA, MYRIAD-SIM, susceptibility maps, and the MYRIAD-EU software support multi-(hazard-)risk identification as well as the monitoring and forecasting of multi-hazard events across scales. Emerging innovations in artificial intelligence (AI) offer further opportunities. As shown by Reichstein et al. (2025), AI applications can enhance

MHEWS, and our review of AI applications across four different pillars of EW4ALL (Tiggeloloven et al., 2025c) provides critical insights into how they may contribute to the implementation of MHEWS.

In facilitating progress and innovations in knowledge development and for multi-(hazard-)risk, ECRs and practitioners are a key component for its success. **Strengthening opportunities for ECR leadership and empowerment within project structures** is essential for such innovation, collaboration, and impact. Reflections on key enabling factors and examples of

empowerment (Schlumberger et al., 2025d) highlight the importance of fostering connection amongst ECRs to enhance integration across the consortium and create pathways for future collaboration, which generates impact both within the project and beyond through strengthening networks, advancing methodological innovations, and facilitating project recognition. Structural inclusion of ECRs, for example through involvement of ECRs in the project management team and establishment of an Early Career Researchers Board, provides a space for sharing of needs and opinions in regard to project

decision making. The experiences within MYRIAD-EU demonstrate that integrating ECR empowerment and encouragement



into the structure, activities, and networks fosters lasting benefits for both the ECRs, the projects themselves, and the broader research community.

**Code availability**. No code was used in this article.

**Data availability**. No data sets were used in this article.

**Author contributions**. PJW coordinated and led the writing of the paper. PJW, SLB, RC, JNC, JD, KDP, MD, SG, SH-S, RST, JS, TT, ST, NvM, and AW (co-led) the writing of various (sub-)sections. All authors contributed to the conceptualisation of the paper, to discussions on the content, and the writing and editing of the paper.

**Competing interests**. Some authors are members of the editorial board of Natural Hazards and Earth System Sciences.

**Acknowledgements**. We extend gratitude to all members of the community who have contributed to both formal and
informal discussion on the topic of multi-(hazard-)risk assessment and management. We thank Eric Mortensen and Lars Tierolf for their contributions to this paper. MYRIAD-EU received funding from the European Union's Horizon 2020 research and innovation programme (grant agreement no. 101003276). The work reflects only the author's view and that the agency is not responsible for any use that may be made of the information it contains.

**Financial support**. This research has been supported by the European Union's Horizon 2020 research and innovation
programme (grant no. 101003276).

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
