# Peer review of "Reducing Risk Together: moving towards a more holistic approach to multi-(hazard-)risk assessment and management"

_EGUsphere, 2025_

## Referee Comment (RC2)

**Review of – Reducing Risk Together: moving towards a more holistic approach to multi-(hazard-)risk assessment and management**

By Philip J. Ward, Sophie Buijs, Roxana Ciurean, Judith Claassen, James Daniell, Kelley De Polt, Melanie Duncan, Stefania Gottardo, Stefan Hochrainer-Stigler, Robert Šakić Trogrlić, Julius Schlumberger, Timothy Tiggeloven, Silvia Torresan, Nicole van Maanen, Andrew Warren, Carmen D. Álvarez-Albelo, Vanessa Banks, Benjamin Blanz, Veronica Casartelli, Jordan Correa González, Julia Crummy, Anne Sophie Daloz, Marleen C. de Ruiter, Juan José Díaz-Hernández, Jaime Díaz-Pacheco, Pedro Dorta Antequera, Davide Ferrario, Sara García-González, Joel Gill, Raúl Hernández-Martín, Wiebke Jäger, Abel López-Díez, Lin Ma, Jaroslav Mysiak, Diep Ngoc Nguyen, Noemi Padrón Fumero, Eva-Cristina Petrescu, Karina Reiter, Jana Sillmann, Lara Smale

In this manuscript, the authors compile the key findings of the recent MYRIAD-EU project and discuss the ways in which this improved our understanding and treatment of multihazard and multi-risk processes. The manuscript is well written and provides an appropriate level of detail – with well-placed citations throughout. The project's scope and achievements are impressive, and this paper will serve as a useful and accessible overview to a broader community. In some places it would be useful to clarify whether the manuscript serves primarily as a summary of key project outcomes, or rather a general commentary on the state of multihazard/multi-risk assessments and management with an evidence base from the project, or a combination of both of these. Where the manuscript aims for more general conclusions, brief discussion of how insights derived from a predominantly European set of case studies may or may not transfer to other global contexts would strengthen the paper. Overall, minor revisions should be sufficient to address these points.

I include line by line comments below:

Title: I am sure there has been much discussion on this point, but is the term '*multi-(hazard-)risk*' the best to use? The complex double hyphen and brackets formulation leaves it open to some confusion, particularly as I am not sure this exact term is explicitly discussed.

Could this be changed from '*multi-(hazard-)risk*' to either '*multi-hazard and multi-risk*' or simply '*multi-risk*'.

To dig further I went into the MYRIAD "D1.2 Handbook of Multi-hazard, MultiRisk Definitions and Concepts" which doesn't have this exact formulation, and different definitions for "Multi-hazard risk" and "Multi-risk" with the difference being that the latter incorporates interrelationships on the vulnerability level. Given the discussion around vulnerability in this article perhaps "Multi-risk" is the most appropriate term? Either way some explicit definition and clarification would help and improve the readability of the article to broader audiences.

Affils – 2. Is just an address. Should this be Deltares?

L37-38 This line "how multi-(hazard-)risk both shapes, and is shaped by, risk dynamics over space and time" is not clear – could it be reworded?

48 'recent mid-term review' -> '2023 mid-term review'

125-130 Is there a mechanism for broader input into this gateway (e.g. 'wiki discussion' type, or comment section type)?

Figure 1- figure is low resolution – can you include the full res version? I can see it on the dashboard (https://dashboard.myriadproject.eu/).

L228-229 Perhaps reword these lines, e.g. neural networks are a type of machine learning.

L237 (paragraph) It would be good to comment on some of the limitations of these ML and data-driven approaches, particularly in areas where data is sparse.

L275 It would be interesting to consider possible limitations here – being based on historical events this sounds like it might work less well in areas with less complete disaster records.

L545 onwards – In this section it might be useful to separate out the notes on what was done in MYRIAD-EU with the outlook recommendations of where multihazard research might go next. There are some excellent points in this section, but it currently feels quite closely tied to what was done in the project, and some changes might open it up some more – for instance, what aspects were not done in this project but would be valuable for future work?

For instance, one aspect that could be discussed is how moving from the European focus of MYRIAD to a global perspective might change some of the lessons here – for instance in areas with different DRM frameworks, greater role of NGOs, few or incomplete event datasets, how might the key points change?

Overall, as an EU project, MYRIAD-EU's team and case studies are focused on Europe, and many key findings have been evaluated within this context. It would be interesting to discuss, throughout the manuscript, which insights are likely to transfer well globally and which may require adaptation outside Europe.

I thank the authors for their contribution and look forward to reading the final paper.

-Max